# Machine Learning and Eye Movements Give Insights into Neurodegenerative Disease Mechanisms

**DOI:** 10.3390/s23042145

**Published:** 2023-02-14

**Authors:** Andrzej W. Przybyszewski, Albert Śledzianowski, Artur Chudzik, Stanisław Szlufik, Dariusz Koziorowski

**Affiliations:** 1Polish-Japanese Academy of Information Technology, The Faculty of Information Technology, 86 Koszykowa Street, 02-008 Warsaw, Poland; 2Department of Neurology, University of Massachusetts Medical School, 65 Lake Avenue, Worcester, MA 01655, USA; 3Department of Neurology, Faculty of Health Science, Medical University of Warsaw, 8 Kondratowicza Street, 03-242 Warsaw, Poland

**Keywords:** Alzheimer’s disease, Parkinson’s disease, eye movements, Rough Set, machine learning

## Abstract

Humans are a vision-dominated species; what we perceive depends on where we look. Therefore, eye movements (EMs) are essential to our interactions with the environment, and experimental findings show EMs are affected in neurodegenerative disorders (ND). This could be a reason for some cognitive and movement disorders in ND. Therefore, we aim to establish whether changes in EM-evoked responses can tell us about the progression of ND, such as Alzheimer’s (AD) and Parkinson’s diseases (PD), in different stages. In the present review, we have analyzed the results of psychological, neurological, and EM (saccades, antisaccades, pursuit) tests to predict disease progression with machine learning (ML) methods. Thanks to ML algorithms, from the high-dimensional parameter space, we were able to find significant EM changes related to ND symptoms that gave us insights into ND mechanisms. The predictive algorithms described use various approaches, including granular computing, Naive Bayes, Decision Trees/Tables, logistic regression, C-/Linear SVC, KNC, and Random Forest. We demonstrated that EM is a robust biomarker for assessing symptom progression in PD and AD. There are navigation problems in 3D space in both diseases. Consequently, we investigated EM experiments in the virtual space and how they may help find neurodegeneration-related brain changes, e.g., related to place or/and orientation problems. In conclusion, EM parameters with clinical symptoms are powerful precision instruments that, in addition to their potential for predictions of ND progression with the help of ML, could be used to indicate the different preclinical stages of both diseases.

## 1. Introduction

Neurodegenerative brain changes start about two decades before the first detectable symptoms [1,2]. During this period, everyone develops various plastic compensatory brain mechanisms. The rates and processes of neurodegenerative disease (ND) progressions have a vast patient-specific spectrum. The main aim of this review is to look, from this perspective, at the early processes related to neurodegenerative changes and to precisely characterize them by using granular computing (GC) or other ML methods. Our leading candidate for a possible early biomarker is related to eye movements (EM). We have concentrated on reflexive EM, such as reflexive saccades (RS), antisaccades (AS), and pursuit EM. Generally, these movements slow down and become less precise with aging, but these changes are more pronounced due to ND. It means that one needs to differentiate between processes related to aging and those related to ND, which is one of the major problems with early ND biomarkers. We base our approach on identifying similarities between different groups of patients and whether patients’ EM and symptoms become more like those of patients in the advanced stages of ND with disease progression. We have demonstrated this approach in different groups of Parkinson’s disease (PD) patients. In the next step, we assumed that more minor cognitive and motor symptoms might be related to more minor changes in EM parameters. This could be a possible approach for the early detection of preclinical ND.

There are three kinds of symptoms in neurodegenerative disease. As different symptoms are related to different structures in the brain, there is an essential question as to whether ND processes are independent of or specific to different parts of the brain. The three kinds of symptoms are as follows:-Cognitive symptoms are dominant in Alzheimer’s disease (AD) but are secondary in Parkinson’s disease (PD);-Motor symptoms are characteristic of PD and less evident for AD;-Emotional symptoms are mostly related to depression and are observed in both diseases but are characteristic of LOAD (late-onset AD) [3,4].

It is widely recognized that neurodegenerative processes start from the basal ganglia in PD and typically from the hippocampus in AD. In PD, the structure that degenerates first is the substantia nigra (part of the basal ganglia), whose neurons are responsible for releasing dopamine. The lack of dopamine slows down and affects movement regulation (GO—NOGO antagonism) [5]. In addition, there is a connection between the striatum and the prefrontal cortex that influences memory and cognition [5]. Finally, as dopamine is the reward neurotransmitter, it affects the reward systems (pleasure feelings) and might lead to depression [3].

The place cells in the hippocampus are responsible for orientation deficiency (navigation problems) in AD, and connections between the hippocampus and the frontal cortex are responsible for memory and cognitive problems. Through its connections to the striatum, the prefrontal cortex might also affect movement in AD. Depression is the main reason for the late onset of AD (LOAD) in older subjects (over 65 years of age) [4].

This review is mainly based on work with Parkinson’s patients under different therapies and at various stages of the disease. We have demonstrated how different ML algorithms can help to predict disease development and how, by using ML, we can compare PD symptoms of less and more advanced patients. Therefore, the main question is: if we can predict disease development, can we also predict the start of ND disease? This question might also be related to looking for sensitive biomarkers that might indicate neurodegenerative brain changes even before the first observable symptoms. We have extensively tested and evaluated eye movement (EM) as a potential biomarker in PD.

Multiple studies have demonstrated that EM helps to predict peripheral movement disorders as well as cognitive and emotional symptom progressions. EM is also affected in AD, so it could be an excellent biomarker for both diseases. One very characteristic parameter of EM in both PD and AD is the delay of the reflexive saccades (RSLat—the time difference between the light spot position change and the start of eye movement). However, RSLat is also related to aging and, to be a significant biomarker, must also be linked to changes in many other, still unidentified, parameters.

To find practical consequences of changes in EM and other ND-related parameters for everyday life, we can immerse the user in a virtual reality (VR) world. The advantage of VR in comparison to the real environment is that one can control significant features of the VR world and test the influence of different parameters on the subject’s behavior. 

Hence, one needs to test a high-dimensional and noisy parameter space, and we suggest that ML methods are the right tool for it.

## 2. Eye Movements and Neurodegenerative Diseases

### 2.1. Standard Neurological Approach

Experienced neurologists use their clinical knowledge and experience from many years of practice to estimate symptom development and the best treatment for an individual Parkinson’s disease (PD) patient. However, because of the long period of compensatory mechanisms unique to each patient, there is a famous saying that “No two people face Parkinson’s in quite the same way.” Therefore, a neurologist must consider not only motor symptoms, but many others, and must even try to understand a patient’s cognitive and emotional symptoms through the Theory of Mind space [6,7] to estimate disease progression in the individual patient.

Typically, eye movement analysis coexists in the broader context of other neuropsychological measurements. They can be related to the following tests: PDQ-39/PDQ-8—a 39/8-question test related to health difficulties in everyday living (summaries of both tests are strongly correlated), ESS—Epworth sleeping scale results (related to sleepiness problems, predominantly during the day), BDI—Beck’s depression inventory (21-item quantitative measure of symptoms of depression), TMT A and TMT B tests (Trail Making Test—part A measures psychomotor speed and part B is related to executive function). Another test, AIMS—Abnormal Involuntary Movement Score—measures the involuntary movements of patients. Additionally, neurologists can use more wide-ranging cognitive tests, such as MoCA (Montreal cognitive assessment) for the detection of MCI (mild cognitive impairment) for PD and AD or dementia (AD), or the similar, but shorter, MMSE (Mini-Mental State Examination), which is less sensitive than MoCA, but has established clinical values. Another important test, the FAS (Phonemic Verbal Fluency) test involves orally producing words that start with the letters F, A, and S, evaluates the cognitive function and measures language-related executive functioning. The most common decisive attribute is the UPDRS (Unified Parkinson’s Disease Rating Scale), an essential neurological test for the effects of PD long-treatment effects. The UPDRS score estimates daily activity (non-motor—UPDRS I and motor—UPDRS II) and motor-related problems (UPDRS III). It is the so-called “Gold Standard” for determining PD progressions. Alzheimer’s disease progression is mainly related to cognitive changes (normal, MCI, dementia), and disease progression is determined by the CDR (Cognitive Dementia Rating) scale as the decision attribute.

### 2.2. EM in PD—Saccades

There is pervasive, clinically oriented, literature related to reflexive saccade (RS) latency in PD in comparison to same-age healthy controls showing different, contradictory results [8]. However, a meta-analytic review [9] demonstrated, by analyzing 47 representative studies (from 1529 references), that the RS latency depends on which method was used by different authors: gap, step, or overlap. Different methods are determined by the time difference (∆t) between the fixation point disappearance and the target appearance: if ∆t > 0, then it is the gap method; if ∆t = 0, then it is the step method; and if ∆t < 0, it is the overlap method. By using quantitative pooling analysis, Chambers and Prescott [9] demonstrated that the slowed response time in PD, compared to a control, is strongest in the step method (∆t = 0), weaker in the gap method (∆t > 0), and negligible in the overlap method (∆t < 0).

In a study [10], standard neurological attributes and PDQ39, Epworth, and AIMS were measured, as well as EM—reflexive saccades (Figure 1). The authors compared different algorithms for the classification of UPDRS, UPDRS II, and UPDRS III in 10 PD patients. A six-fold cross-validation method was used. For the accuracy measure of UPDRS, the best results were given by RSES (Rough Set Exploration System based on Rough Set theory, which is an important implementation of GC—granular computing) with a global accuracy of 0.90, the second was Random Forest, which had a global accuracy of 0.68, and the third was a Tree Ensemble, with 0.65 global accuracy. For UPDRS II, the best results were for the Random Forest with an accuracy of 0.80, the second was RSES with 0.79, and the third was Bayesian classification, which gave an accuracy of 0.77. For the UPDRS III, the best was RSES classification, which gave a 0.82 accuracy, and the Decision Table (with Weka), which gave an accuracy of 0.77.

The results above agree in general with a recent publication [12] where they tested horizontal and vertical saccades and antisaccades (AS) for healthy control and PD subjects in different stages of the disease, measured by Hoehn and Yahr stages (H&Y2 and H&Y3). The PD group displayed decreased vertical saccade amplitudes and increased vertical saccade and AS latencies. AS latency increased for H&Y2 and H&Y3 patients, but AS errors (correlated with MoCA—Montreal cognitive assessment test’s score) were similar for control and H&Y2 subjects, but larger for H&Y3 subjects. Levodopa has increased vertical saccade latency but decreased AS latency. This work has described many different difficult-to-analyze mechanisms that might be easier to put through machine learning methods.

In one study [13], it was demonstrated that saccades could predict cognitive decline in PD patients. In 140 PD and 90 age-matched participants, the authors evaluated differences in RS metrics between early-PD and healthy age-matched adults. They assessed RS and cognition at baseline at 18, 36, and 54 months. RS parameters were latency, duration, amplitude, peak velocity, and average velocity, and the cognitive assessment contained executive function, attention, fluctuating attention, and memory. RS parameters, with the help of linear mixed-effects models, were used as predictors of cognitive decline over 54 months. At the baseline, RS was impaired in PD patients compared to the control group. RS parameters predicted a decline in global cognition, executive function (verbal fluency), attention, and memory over 54 months in PD patients. However, only reductions in global cognition and attention were predicted by RS parameters in age-matched subjects, which means that cognitive changes were not just age-related [13]. The dependence between RS latency and executive functions was also confirmed earlier [14]. In addition, manual and saccadic performances are uncorrelated in the average population, but both are similarly affected by PD [15].

Abasi et al. [16] tested whether vestibular therapy exercises might recover both oculomotor functions and postural control (in the upright position) in patients with PD. Vestibular therapy is a set of exercises that affect the basic elements of central sensorimotor integration. They tested 11 idiopathic PD patients voluntarily contributing to the survey based on the following criteria: central vestibular dysfunction and Hoehn and Yahr scale scores ≤3. Videonystagmography (VNG) and Berg Balance Scale (BBS) scores were measured as the control. PD patients undertook vestibular rehabilitation training for 24 sessions (3 sessions per week). VNG and BBS scores were appraised again 48 h after completing the last session of the exercise. Investigators detected significant advances in balance (p≤0.015), and eye-tracking and gaze performance were statistically enhanced in seven and six patients, respectively.

Wong et al. [17] investigated the relationship between EM parameters and execution in different cognitive tests in Parkinson’s patients. In the eye-tracking experiment, subjects were asked to look for a number embedded in an array of alphabets distributed randomly on a workstation screen. Researchers calculated the average amplitude of saccades and fixation duration and correlated these data with the results of cognitive tests. It was established that prolonged fixation time was linked with inferior performance in verbal fluency, as well as in visual and verbal memory, showing that EM parameters are alternate markers for cognitive function in PD patients.

Wong et al. [18] studied the correlations between EM and cognitive functions in patients with Parkinson’s disease. A total of 62 patients with non-demented Parkinson’s disease and 62 controls of the same sex, similar age, and equivalent education were exposed to cognitive and oculomotor tests. Researchers observed a negative correlation between the length of eye fixation and functioning in semantic verbal fluency, as well as verbal and visual memory undertakings. Researchers concluded that increased duration of visual fixation correlates with poor results in semantic verbal fluency and verbal and visual memory tasks in non-dementia PD. In another 2-year follow-up study, 49 of the primary 62 patients (15 cases/31% were classified as MCI (with mild cognitive impairment)) were examined in the context of the relationship between domain-specific cognitive impairment and the progression of visual fixation duration. Researchers noticed that the duration of fixations increased significantly after two years. For the analysis, they used ANOVA with repeated measurements, and according to the results, impairment of semantic verbal fluency, visual and verbal recognition memory, and indicative attention function had a significant effect on the extension of the duration of visual fixation [18]. Repeated measures beyond two years showed a correlation between prolonged visual fixation and different types of cognitive impairment associated with cholinergic dysfunction. According to the authors, it provides preliminary evidence that the observed eye-tracking model is a surrogate marker for the cholinergic deficit in Parkinson’s disease.

Archibald et al. [19] assessed the error rates and visual exploration tactics of subjects with Parkinson’s disease, in relation to the extension of their cognitive impairment, while performing a battery of visuo-cognitive tasks. They found that error rates were significantly higher in those PD groups with either MCI (*p* = 0.001) or dementia (*p* < 0.001) in comparison to cognitively normal subjects with Parkinson’s. When matched to cognitively normal Parkinson’s disease patients, the exploration tactic, as measured by EM-tracking variables, was least efficient in the dementia group and inefficient to a lesser extent in MCI patients. In control subjects and cognitively normal subjects with Parkinson’s, it was established that saccade amplitude was drastically reduced in the groups with MCI or dementia.

The fixation period was stretched in all PD patients in relation to healthy control subjects but was most prolonged for cognitively impaired PD groups. The average fixation period was strongly related to disease severity. The authors concluded that the increase in the fixation period, existing even in cognitively normal patients with PD, implies a disease-specific influence on the systems directing visual search.

### 2.3. EM in PD—Antisaccades

One of the well-proven experimental models used to examine the inhibition of automatic reflexive responses is the antisaccade task (AS) [20].

In a study [21], the significance of antisaccade (AS) parameters for the classification of Parkinson’s disease motor and motor variations (UPDRS II and UPDRS IV) was tested. There were 11 PD patients examined in 4 sessions. In addition to the standard neurological attributes, AS parameters such as delay, duration, and maximum speed were measured. RSES was used for the data discretization and attribute reduction and to perform a 5-fold cross-validation. The best result was obtained by the RSES Decomposition Tree, which splits the dataset into fragments represented as a tree’s leaves [21]. The UPDRS III classification results indicated an accuracy of 0.85 with a coverage of 0.48. Surprisingly, the UPDRS IV was estimated with an accuracy of 0.91 and coverage of 0.39, so UPDRS IV showed a more significant correlation with antisaccade parameters. Thus, UPDRS IV showed greater sensitivity in predicting antisaccade parameters [21]. From the results, it also emerged that attributes describing methods of patient treatment (again, the session attribute) and mean duration were most sensitive in predicting the scores of both UPDRS III and IV. An example of AS in two different PD patients is shown in Figure 2.

As described above, different analysis methods influence the RS latency [9]. In a review meta-analysis, Waldthaler et al. [22] analyzed the influence of the paradigm (gap, step, overlap) on AS latency and errors. They [22] compared the results of 703 PD patients with 600 healthy controls for antisaccade latency and 831 patients and 727 healthy controls for antisaccade error rate. Over 60% of studies excluded PD with dementia. Like RS latencies, the mean AS latency was 339.8 ms in the PD patients and 294.2 ms in the healthy group in the gap paradigm, and 411.7 ms in the PD patients and 368.6 ms in the healthy group in the step paradigm. This was measured for PD patients with disease duration between 0.7 and 14.7 years and UPDRS III scores between 5 and 85, from early to advanced disease stages. In a meta-analysis, the authors [22] demonstrated that AS latency increases with disease severity, and an increase in the levodopa dosage influences the AS error rate (negatively moderating effect).

A study by Waldthaler et al. [23] tested whether patients with Parkinson’s taking dopaminergic medication performed better at response inhibition during antisaccade tasks. Levodopa intake has favorable or harmful effects on dopamine-dependent cognitive tasks based on essential basal dopamine intensities in ventral segments of the striatum, agreeing with the dopamine overdose theory. Thirty-five patients with Parkinson’s (and 30 healthy subjects) completed antisaccade tasks in OFF and ON medication conditions. Investigators computed multiple linear regressions to forecast the alterations in antisaccade delay and directive mistakes, and to express saccade rate based on age at Parkinson’s disease onset, disease duration, levodopa-equivalent circadian amount, motor indicator difficulties, and executive functions. According to their results, earlier disease onset and milder motor symptoms in the OFF-medication status were related to diminished inhibition ability response after levodopa intake, mirrored in enlarged express saccades and mistakes. They concluded that levodopa might have opposite results on oculomotor reaction inhibition contingent on the age at Parkinson’s disease onset and motor disease gravity.

During their next study, Waldthaler et al. [24] examined whether there was any correlation between the development of motor and cognitive indications in 25 patients and Parkinson’s disease (age: 61.4 +/− 6.8, disease duration: 6.0 +/− 4.5 years). A total of 10 patients from all 25 PD patients received subthalamic nucleus DBS (deep brain stimulation) during the follow-up period (from DBS surgery to follow-up visit: 4.5 +/− 2.1 months). All PD patients were examined in ON medication and ON-DBS states, and modifications of dopaminergic treatment were permitted during the follow-up epoch. PD patients without DBS who displayed substantial improvement in motor signs after one year also received higher levodopa equivalent dosages at follow-up. Generally, the antisaccade (AS) delay (baseline: 339 +/− 72 ms, mean change: 95 +/− 1.1 ms) and mistake rate (baseline: 0.52, mean change: −0.02 +/− 0.3) stayed steady in the non-DBS group. In the DBS group, the AS delay tended to increase (baseline: 295 +/− 78 ms, mean change: 48 +/− 75 ms (*p* = 0.09)), but the mistake rate improved at follow-up (baseline: 0.76, mean change: −0.21 +/− 0.3 (*p* = 0.048)). The change in AS delay was connected to change in MDS-UPDRS III in both groups (non-DBS group baseline: 25.7 +/− 13, mean change: 0.3 +/− 7.4; DBS group baseline: 27.2 +/− 16.5, mean change: −5.2 +/− 17.8) and with the change in MoCA score in the non-DBS group (25.8 +/− 3.1, mean change 1.3 +/− 3.1). The authors indicate that AS delay may be sensitive to the development of motor and cognitive signs over time in Parkinson’s disease patients.

### 2.4. EM in PD—Saccades and Antisaccades

The same group of patients as in [21] was used for UPDRS prediction based on RS and AS measurements. The best accuracy of 0.89 was achieved by Decision Trees [11]. The results showed that the accuracy of the predictions increased with the number of significant attributes that were obtained by, for example, averaging RS and AS duration or by adding the averaged standard deviations of each patient’s latencies [11].

The authors of [22] demonstrated that RS and AS latencies were correlated with the results of neuropsychological tests in 65 PD patients, but only the results for AS latencies concerning patients’ cognitive impairment were statistically significant. In a study [25], 19 drug-naïve PD patients and 20 age-matched controls were examined. Patients had clinically probable idiopathic disease within three years of disease onset. Their RS latencies were like those of the controls, but AS error rates differed significantly (PD 15% vs. 8.7% for controls).

Fooken et al. [26] studied different tasks and conditions in which the oculomotor function in Parkinson’s patients is preserved. A total of 16 patients with Parkinson’s disease and 18 healthy, age-matched controls performed a set of tasks of saccades (RS), anti-saccades (AS), pursuits, and rapid ‘go/no-go’ manual interventions. Compared to the control group, PD patients showed regular impairment in tasks with fixed targets: prosaccades were hypometric, and AS were wrongly started towards the indicated target in 35% of the trials compared to 14% of errors in the control group. In PD subjects, task errors were linked with short-latency saccades, demonstrating anomalies in inhibitory control. However, the patients’ EMs in response to dynamic targets were well-preserved. Parkinson’s disease patients can track and predict a moving target and make quick go/no-go decisions with the same precision as healthy people. The intercepting hand movements of the patients were slower on average but indicated adaptive processes compensating for the motor slow down. Researchers concluded that the preservation of eye and hand movement functions in PD is linked to a separate functional pathway through the upper colliculus–brainstem loop that detours the frontal–basal ganglia network.

Kocoglu et al. [27] investigated how social processes and behaviors change in PD during spatial signaling tasks. Socially relevant directional cues, such as photos of people looking left or right, have been found to redirect attention. In conclusion, the basal ganglia can play a role in responding to such directional signals. In this research, patients and healthy controls performed pro- and anti-saccade tasks in which different directional signs preceded the appearance of the target. They analyzed reaction time, prediction errors, and correlations with PD severity and cognitive assessment scores. Patients displayed increased errors and answer times with the AS (antisaccade) task, but not with the RS (saccade) task. The control subjects made the most predictive errors in the finger-pointing trials, and the PD patients were mostly affected by the arrow, gaze, and pointing clues. It has been found that PD patients have a reduced ability to suppress responses to directional signals, but this effect is not specific to social signals.

Munoz et al. [28] studied whether bilateral deep stimulation of the basal ganglia–subthalamic nucleus (STN DBS) may affect the control of inhibition of eye movement in PD. They investigated the effect of DBS amplitude on inhibitory power during an antisaccade procedure on 10 PD patients after their DBS surgery. Subjects without medication (12 h, overnight) performed the antisaccade tasks with a set of different DBS stimulation amplitudes (from 0—no stimulation to 5—higher levels). The prosaccade error rate (related to a saccade at the beginning of the antisaccade) increased with increasing DBS stimulation amplitude (*p* < 0.01). Moreover, the saccade error rate increased with the decrease in the modeled volume of tissue activated (VTA) and decreased overlap of the STN stimulation area, but this connection was determined by the stimulation amplitude (*p* = 0.04). They concluded that the directional prosaccade error rate during the antisaccade task indicated impaired inhibitory control and suggested that higher stimulation amplitude settings can be modulatory for inhibitory control.

### 2.5. EM in PD—Pursuit

Another study tested how effective diagnostic parameters of slow (pursuit) eye movements are for the prediction of PD symptom development [29,30]. Horizontal pursuit EM with three different sinusoidal movement speeds was measured. The gain and accuracy (EM measurement section) were estimated. The discretization and attribute reduction with RSES demonstrated that the significant attributes were precise for the accuracy of the fastest sinusoidal movement speed, and gains decreased for medium and high sinusoidal movement light spot speeds [29,30]. The result of the 4-fold cross-validation gave a global accuracy of 0.77 for the UPDRS III prediction. An accuracy of 0.8 for the session number prediction (different treatments) in 10 PD patients was found. The above predictions were obtained for a sample of 20 patients using different binning methods (KNIME auto-binner), which allowed the grouping of UPDRS III data in intervals of equal frequencies. A 90% accuracy in predictions on these data was achieved with the RSES and 5-fold cross-validation [30]. When comparing the accuracy results of different classifiers, the RSES is in first place in the ranking, ahead of SVM (59%), Naive Bayes (55%), and Random Forest (52%) [30]. 

In this context, in her review, Frei [31] analyzed 29 articles (from 819 found) on smooth-pursuit eye movements in PD patients and compared them to those in normal subjects. She found that in 18 articles, the gain was measured and reduced in PD patients compared to controls in 16 of these papers. In two papers, the gain was reduced for higher target velocities. In three articles, accuracy was measured and found to be reduced in PD. There were also correcting saccades during smooth-pursuit EM that were more dominant in more advanced PD and for faster smooth pursuits, but quantification of saccades was difficult [31].

In another study, deep brain stimulation (DBS) increased smooth-pursuit accuracy (*p <* 0.001) and smooth-pursuit gain (*p* = 0.005), especially for faster smooth pursuits (*p* = 0.034) [32].

In their study, Farashi et al. [33] observed eye movements (EMs) during inactive states (eyes closed and eyes open), measuring EM using vertical electrooculography (VEOG). They performed the analysis in the time, frequency, and time–frequency axes of the VEOG time series. The authors completed a categorization by comparing healthy subjects and PD patients in OFF and ON medication conditions. They used an SVM (support vector machine) classifier and allowed multiple-differentiation-corrected *p*-values. The VEOG data achieved 69.10% and 87.27% discrimination precision for OFF and ON medication conditions, respectively. The authors established that PD patients’ vertical EM had smaller amplitude changes than healthy subjects in OFF medication conditions. The levodopa treatment augmented such changes in vertical EM during the eyes-closed situation and diminished during the eyes-open situation. As a result of levodopa treatment, VEOG time series amplitudes may change, although vertical EM rates were not affected (frequency contents).

### 2.6. EM in PD—Pupillometry

Parkinson’s disease patients develop a distorted pupillary response dependent on an abnormality in the retinal ganglion cells. Tabashum et al. [34] illustrated an arrangement for pupil size estimates that permits the discovery of pupil parameters to measure the post-illumination pupillary response (PIPR) with a Kalman filter estimating the pupil center and diameter over time. The pupillary reaction was estimated in the contralateral eye to two diverse light stimuli (470 and 610 nm) for 19 Parkinson’s patients and 10 healthy subjects. Net PIPR displayed different reactions to wavelengths (0.13 mm for Parkinson’s patients and 0.61 mm for healthy subjects, proving an extremely significant differentiation (*p* < 0.001)).

Tsitsi et al. [35] evaluated gaze constancy and pupil size in steady light surroundings, as well as eye movements (EMs) during constant fixation in a group of 50 Parkinson’s disease subjects (66% males) with unilateral to mild symptoms (Hoehn and Yahr 1–3; Schwab and England 70–90%) and 43 control subjects (37% males) with an eye tracker (1200 Hz) and logistic regression analysis. They examined the potency of the relationship of EM measures with the ROC curve results of 0.817, 95% CI: 0.732–0.901, and concluded that eye-tracking-established amounts of gaze fixation and pupil reaction might be valuable biomarkers of Parkinson’s disease indications.

### 2.7. EM in PD—Multimodal Approach

Bonnet et al. [36] investigated how connections between vision and posture are exaggerated in Parkinson’s patients. PD subjects have been shown to display unusually low levels of synergy in their posture self-control. These impaired reactions are related to the neurodegeneration processes in Parkinson’s disease that affect the basal ganglia, which facilitate the integration of both types of movements. They tested 20 PD patients (mean age: 60) on levodopa and 20 age-matched-healthy subjects (mean age: 61) with a detailed visual assignment (target-seeking scenarios in an image) and an inaccurate control task (arbitrarily viewing an image) in which pictures were projected onto a large screen. Lower back, upper back, head, and EM were registered simultaneously. To analyze behavioral synergies, the authors computed Pearson correlations between EM and postural actions. The associations between EM and upper- and lower-back movements were diminished in Parkinson’s subjects. The healthy control subjects did not display important correlations between EM and postural activities. Generally, their results revealed that the Parkinson’s subjects were unable to correct and change their postural rigidity to achieve success in the visual task. Moreover, these problems may occur in the early stages of Parkinson’s (an early biomarker opportunity).

Zhang et al. [37] investigated 49 Parkinson’s patients, including 35 early-stage (Hoehn and Yahr: 1–2 staging) and 14 advanced PD subjects (Hoehn and Yahr scale: 3 to 5 staging) and 23 healthy subjects. In addition to clinically significant PD symptoms, video-oculography was used to measure EM features such as eye fixation stability, horizontal and vertical reflexive saccade (RS), and horizontal and vertical smooth-pursuit movements. The authors discovered that five EM features—specifically square wave jerk frequency, vertical RS delays, the accuracy of the vertical–upward RS, and the horizontal smooth-pursuit RS gain—were meaningfully different in Parkinson’s and normal subjects. By merging all five features, the authors achieved a symptomatic sensitivity of 78.3% and a specificity of 95.2%. The study discovered that more deficiencies in upward–vertical RS than in other directions were related to disease duration and the stage of development of Parkinson’s disease.

Perkins et al. [38] investigated whether Sleep Behavior Disorder (RBD) indicates PD. With video-based eye tracking, researchers tested saccade, pupillary, and blink responses in RBD and isolated REM (rapid eye movement) with 22 PD and 22 RBD patients and 74 healthy controls. They found that RBD patients did not have significantly different saccades compared to healthy controls, but PD patients differed from both healthy controls and RBD patients. They concluded that RBD and PD patients had altered pupil and blink behavior compared to healthy controls. Because RBD saccade parameters were comparable to healthy controls, brain areas responsible for pupil and blink control may be impacted before saccadic control areas, making them a potential prodrome of PD.

### 2.8. Prediction of Disease Progression in Different PD Groups

The goal in [39] was to predict Parkinson’s disease progression in advanced-stage patients based on data obtained from patients under different treatments and at different stages of the disease. Patients from the BMT group (only on medication, third visit), DBS group (after recent deep brain stimulation surgery, third visit), and POP group (after older DBS surgery, first visit) were used as a training dataset—a model. The model was tested on the data obtained from the POP group during the second visit. A dedicated data science framework written in Python was used and based on the Scikit Learn and Pandas libraries that implemented different multiclass strategies, such as k-Nearest Neighbors Classifier, Support Vector Classifier, Decision Tree Classifier, and Random Forest Classifier. In this trial, the Random Forest Classifier achieved the highest overall accuracy score of 0.75 and an accuracy of 0.7 when predicting subclasses of UPDRS for patients in advanced stages of the disease who responded to treatment, with a global 0.57 accuracy score for all classes [39].

The purpose of another study [40] was to predict the results of different PD patient treatments to find the optimal one. The study compared the intelligent methods based on Rough Set theory with several different machine learning algorithms, namely Gaussian Naive Bayes, Decision Tree, Logistic Regression, C-Support Vector, Linear SVC, and Random Forest. Generally, the Rough Set method gave better accuracy, but less coverage, than other algorithms. On the other hand, the Rough Set-based approach allows the creation of more general rules without the necessity of additional data splitting (into different sessions), which was required in the other ML models to obtain accuracies similar to those obtained by RS. An example is the prediction of UPDRS in a DBS patient group from rules obtained from BMT patients. Global accuracy for DBS patients was 0.64 for the first visit, 0.85 for the second visit, and 0.74 for the third visit. Other methods gave accuracies of 0.88, 0.58, and 0.54, respectively [40].

The principal conclusion from this comparison is the observation that RS is a much more universal method when considering medical data. Finally, it was demonstrated that it is possible to estimate symptoms and their time development in populations treated differently, which may, in the future, lead to the discovery of universal rules of PD progression and to the optimization of treatment.

### 2.9. Prediction of Disease Progression Related to Motor, Cognitive, and Emotional Longitudinal Changes in PD Patients

In [41], two BMT groups of patients (only on medication) were tested. The first one, less advanced, was tested three times every half year (visit 1, visit 2, visit 3). In the second BMT group, more advanced patients were tested only once. All tests were performed with the following condition attributes: PDQ39, Epworth, depression score (Beck test), TMT A and B, disease duration, and fast EM. The decision attribute was UPDRS. With the help of Rough Set theory (RSES), rules describing the more advanced BMT group were constructed and used to predict disease progression over three visits in the less advanced BMT group of patients. Using all condition attributes, general rules gave accuracies as follows: visit 1—0.68, visit 2—0.86, and visit 3—0.88. When rules were related only to motor attributes, the accuracies were as follows: visits 1—0.80, 2—0.93, and 3—1.0. For rules related to cognitive attributes, the results were as follows: visit 1—0.50, visit 2—0.60, and visit 3—0.64. The higher accuracy can be interpreted as more similar patient symptoms. General and motor-related accuracies increased with disease progression (visit numbers), which means that the less advanced group of patients became more like the advanced group. However, this was not the case for cognition-related symptoms that gave lower accuracies, which means that their progressions were not as strongly correlated with disease development.

The influence of the patient’s emotions on the accuracy of the predictions of disease progression in the same group or different groups of patients was also tested through the depression score (Beck test) [42]. The progressions of the BMT group (only on medication) for visits 2 and 3 and the DBS group (deep brain stimulation) for visit 1 were compared based on the BMT symptoms during visit 1. The predictions were performed with the help of RSES and with standard neurological testing and EM parameters. Based on rules from first visit BMT patients, the prediction of symptoms (UPDRS) of BMT for visits 2 and 3 had accuracies of 0.7 and 0.7, but by adding the depression score, accuracies increased to 0.77 and 0.80 [42]. Similar predictions were calculated for the DBS group progression based on first visit BMT rules. Accuracies obtained for the DBS group were as follows: visit 1—0.64, visit 2—0.77, and visit 3—0.74. Adding the depression score to all attributes, improved accuracies of visit 1 to 0.77, visit 2 to 0.85, and visit 3 to 0.8 were demonstrated [42]. In summary, the depression score has a significant influence on predicting Parkinson’s disease progression.

### 2.10. EM in AD vs. PD

The impairment of the oculomotor system in AD manifested with longer RS latency along with higher variability in accuracy and speed [43]. Yang et al., 2012, found similarities between three groups: AD patients, patients with amnestic mild cognitive impairment (aMCI), and healthy elderly subjects [43]. All groups showed shorter latencies in the gap tests (when there is a time delay between the disappearance of the fixation spot and the appearance of the light spot in the periphery) than in the overlap tests (when the above spots’ appearance overlaps in time). However, in both tests, AD patients showed abnormally long saccade latencies. Although there was no significant difference in the accuracy (gain) and the velocity (both mean and peak velocity) between the three groups of subjects, AD patients showed an abnormally high coefficient of variation in the latency, accuracy, and speed of the reflexive saccades. There was a significant correlation between scores for the Mini-Mental State Examination (MMSE) and latencies of the saccades when comparing the MCI subjects to healthy elderly subjects [43].

Wilcockson et al. [44] explored AS eye movements in patients with amnestic and non-amnestic variants of MCI. There were 68 patients with dementia due to AD, 42 had amnestic MCI (aMCI), 47 had non-amnestic MCI (naMCI), and 92 were age-matched healthy controls (HC). The latencies for AS correction in the AD group were significantly longer than those for the HC and naMCI groups, but AS latencies in the AD group did not differ significantly from latencies in the aMCI group, even after age difference corrections [44]. They obtained similar results for the percentage of uncorrected AS errors. The AD and aMCI groups had similar and higher error rates than the naMCI and HC groups. This demonstrated that MCI patients are more likely to develop dementia due to AD than age-matched healthy adults. People with aMCI are at the highest risk of progressing to AD [45], and AS measurements might be an additional prognostic tool for predicting which people with MCI are more likely to progress to AD. It is worth noting that AS latency is a sensitive measure of the inhibitory process and is related to disease progression in the early stages of AD and PD.

In research by Pereira et al. [46], MCI sufferers were similarly impaired in their voluntary saccadic reaction times compared to AD sufferers, with a longer time to correct erroneous saccades.

Boxer et al. [47] compared saccade and antisaccade parameters in patients with frontotemporal dementia (FTD), patients with AD, and healthy subjects. The patients with AD showed an increased saccade latency compared to the FTD group during the horizontal saccade tasks. This might be related to the different dorsal parietal lobe roles in these two groups of patients [47]. In the AS task, all FTD and AD patients were impaired relative to the healthy subjects. The AD patients made fewer correct AS than controls, and they had more difficulty correcting saccade direction when they began from saccade instead of AS [46].

The relationships between AS parameters and measures of inhibitory control, attention, working memory, and self-monitoring showed correlations and common patterns reflecting deficits in executive function, confirming cognitive impairment in MCI and AD patients [46].

In Figure 3, we compared the latency of the reflexive saccades for normal subjects with those of AD and PD patients. These are averaged values for patients in different stages of the disease. However, latencies for AD and PD patients look similar and they are significantly longer than the mean RS latency for normal subjects.

Three studies found a reduction in pursuit gain along with an increase in correction saccades in patients with Alzheimer’s disease [48,49,50].

## 3. Further Research

The ultimate research goal is to identify unrecognized changes in the brain which can cause AD and PD. We think introducing the methods to a wider audience might enable the fulfillment of the following points:-Results must be based on a broader control group.-Tests must ensure repeatability and reproducibility in a non-experimental environment.-Methods must be extended with new digital biomarkers that can be observed in a three-dimensional space.

Therefore, our research team aims to design, evaluate, and introduce modern methods of data aggregation based on online self-assessments and virtual reality environments.

### Virtual Reality—A Research Opportunity

In recent years, advances in technology related to display, computation, and controllers have brought to the market new solutions that have changed digital content consumption, including virtual reality (VR) technology.

To classify this approach, Milgram et al. [51] introduced the term “virtuality continuum”, relating to the mixture of classes of objects presented in any display situation. This representation describes a superset of the user’s perception of the environment. The continuum starts from the authentic environments (consisting solely of natural objects) and ends in completely simulated, virtual environments. It includes different stages of representative forms, such as augmented reality (AR) and virtual reality (VR). These representations of the environment and the areas interpolated together comprise the term Extended Reality (XR). VR refers to devices that occlude the user’s view of the physical world only, allowing sight of digitally rendered images. VR devices can mimic stereoscopic vision by presenting slightly different, separate images to both eyes. The main idea is to immerse the user in the virtual world by depriving external stimuli during a content presentation. VR devices are headsets that entirely cover the field of view (FoV) and project the image directly to both eyes.

The wide availability of devices and improving quality mean they are being used on an increasing scale. Therefore, VR technology could be the next gold standard in cognitive assessment. The need for new tools has emerged from criticism of the current cognitive screening tools because these tests often (30 of 50 classical screening tools) miss a visuospatial component, such as the Clock Drawing Test, the Cube Drawing Test, and the Intersecting Shapes Test [52]. Because visuospatial tasks demonstrate significant diagnostic and prognostic potential in AD [53], VR applications have great potential as an assessment tool in dementia [54]. One example is shown in Figure 4, where the standard executive function Trial B test is performed in 3D instead of 2D (on the paper). In this case, in addition to engaging executive processes related to 2D, the subject must also activate 3D orientation processes that often fail in ND (especially in AD).

One of the main concerns of VR usage might be the physiological phenomenon called VR sickness (cybersickness). The most usual explanation is the sensory conflict theory [55]. According to this theory, VR sickness is caused by discrepancies in the sensory communication sent to the brain as the operator progresses through the virtual environment. Symptoms frequently reported include general distress, headache, eyestrain, stomach sensitivity, nausea, sweating, spite disorder (a.k.a. drowsiness), disorientation, and a nausea response. Symptoms can last from minutes to days post-exposure, with after-effects displaying as postural ataxia, visual displacement (e.g., altered vestibular-ocular reflex), and altered hand–eye harmonization, among other disorders. Cybersickness (CS) has been called the “elephant in the room” due to the possibility for it to radically limit VR equipment’s uptake.

There are individual differences in susceptibility to VR sickness, and age is one of the factors. Brooks [56] presented that over-50s are more likely to experience virtual reality sickness than younger adults. Another factor is gender. Park [57] discovered that women are more vulnerable to virtual environments in terms of motion sickness (MS) or simulator sickness (SS). Furthermore, Kennedy [58] claimed that women are more susceptible than men, and the main reasons could be hormone differences and that women have a wider field of view. Depending on the immersive content, 20–95% of users typically experienced some form of cybersickness, ranging from a slight headache to an emetic response. This was not the conclusion in more recent research [59], which found no evidence that the incidence of motion sickness or the severity of motion sickness symptoms differed between the sexes.

These factors are common obstacles that impact the widespread use of technology, particularly among the elderly. However, with improving technology (higher resolution and frequencies), this problem seems likely to fade and should be eliminated in the future. Caserman [60] conducted a meta-analysis and compared different head-mounted displays (HMD) and stimuli. For example, Oculus Rift HMD vs. HTC Vive HMD and matched stimuli vs. unmatched stimuli. The meta-analysis results show that last-generation HMD devices have significantly fewer problems with CS, although they are still present. The findings reveal user experience (UX) flaws that could be obstacles in medical research. Thus, research group selection must be performed carefully and precisely until the technology can deal with cybersickness.

Immersive virtual environments allow researchers to create realistic environments while maintaining a high level of experimental control. For example, Garcia [61] suggests that it is possible to create experimental conditions where a virtual human modifies tone of voice while maintaining neutral facial expressions that would allow the study of the impact of tone of voice on persons with dementia. Furthermore, Flynn [62] presented findings that demonstrated that it is feasible to work in virtual environments with people with dementia. Bek et al. [63] found differences in eye gaze for emotional expressions which are static and dynamic. According to the researchers, PD may reduce the ability to utilize motion in emotion recognition, and eye movements reveal subtle effects of motion on emotion processing in PD. Researchers concluded that measuring eye gaze for moving faces enhances understanding of emotion recognition.

Additionally, as presented in [64,65], early detection of Alzheimer’s disease can be supported by navigation tasks. This study used an immersive virtual reality paradigm in which participants walked through simulated environments to investigate path integration tasks. This study shows that a virtual reality navigation task can distinguish patients with moderate cognitive disabilities at low and high risk of developing dementia with better classification accuracy than classic cognitive measures. Virtual reality spatial cognition assessments have also been shown to be more sensitive than traditional visuospatial pencil-and-paper tests, such as the Mental Rotation Test, in detecting spatial navigation deficiencies [66]. Research papers also present a definite connection between virtual and real-world findings, for example, proof that wayfinding navigation performance on a mobile app-based VR navigation task is closely correlated with real-world city street wayfinding performance [67].

A VR environment also provides vast possibilities for diminished curiosity research, which are behavioral changes that are extremely difficult to measure experimentally. A particular group of studies devoted to novel visual object perception (curiosity) associated with aging [68,69,70,71] presented that AD patients distributed their viewing time equally and spent significantly less time than controls looking at the novel (unpredictable) stimuli versus classical stimuli (in comparison to healthy subjects). Such novelties can be a horse that appears to have no hind legs or a lion that appears in a children’s classroom, as in the classical study from 1992 [68]. Because a VR environment with proper hardware enables the simulation of real-world and synthetic objects with outstanding detail, one can automatically measure the subject’s attention span based on eye movement registration in 3D space for prominent and less obvious examples of artifacts.

Moreover, Mandera [72] shared the opinion that virtual reality can be a supportive therapy for patients with MCI and various forms of dementia to improve adherence to cognitive training of older adults with cognitive impairment. This opinion is coherent with the outcome of the meta-analysis prepared by Kashif et al. [73]. Out of nine studies on motor function, six reported equal improvements in motor function compared to other groups. In addition, VR groups achieved superior results in improving static balance among patients with PD.

Our research group also evaluated the possible impact on early disease detection. We created a prototype of an application testing the recognition of numbers and letters in the correct order implemented in a VR environment. The test assumed that Trail B is generally sensitive to executive functioning since the test requires multiple abilities to complete it. Part B requires attention, memory, visual screening abilities, motor functioning, and cognitive processes in 2D, and we intuit that it is more difficult in 3D.

In the context of previous research, it is worth noting that there are VR headsets with eye tracking on the market, such as the HTC VIVE Pro Eye. Hence, they can be used as standalone research environments that allow us to control the experiment remotely, without on-site supervision. Furthermore, they add a spatial dimension to our research, connected to the motor reaction of the eyes combined with actions executed by the subject’s hands.

## 4. Discussion

In this review, we have demonstrated that the parameters of reflexive eye movement are significant in estimating Parkinson’s and Alzheimer’s disease progressions. Therefore, they might also be good biomarkers in the preclinical stages of these ND diseases. Various saccade abnormalities were found in Parkinson’s [74,75,76,77] and Alzheimer’s diseases, e.g., in reviews [78] and using computational attention models under realistic scenarios for AD [79]. Even if many authors have demonstrated EM pathologies in AD and PD, they did not demonstrate how we can use EM parameters to predict the disease progression of many different patients or even patients with different treatments and symptoms. However, in [80], the authors used LR (logistic regression), SVM (support vector machine), and NB (Naive Bayes) algorithms to classify normal (NC), MCI, and AD subjects based on novelty preference (NP), pupil diameter (PD), saccade orientation (SO), and re-fixation (RE) and fixation duration (FD) differences between watching a familiar or a novel image (see discussion above related to [68,69,70,71]). The division into NC (*n* = 30 subjects), MCI (*n* = 10), and AD (*n* = 20) was assessed by clinicians based on standard assessments and neuropsychological tests [80]. The authors used a cross-validation method on 20 NC and all AD subjects to determine classification algorithms that distinguish between AD and NC. In the next step, they used this algorithm to distinguish NC from MCI. They repeated the classification process 100 times by changing partitions of NC subjects and, each time, testing different NC subjects against all MIC patients and averaging all results. The best results were obtained for all attributes: NP + PD + SO + RE + FD. For the SVM algorithm, accuracy was 0.87 and sensitivity 0.97. These are great results, giving the basis for our proposal for VR testing.

In [81], the authors used oculomotor behavior to differentiate diagnoses between normal subjects, AD sufferers, and behavior variants of frontotemporal dementia (bvFTD) and a semantic variant of primary progressive aphasia (svPPA) groups. They tested RS, AS, and pursuit EM. By comparing RS latency, AS success (successfully performed AS), and spatial accuracies of RS, AS, and pursuit, the authors found that AD patients performed the worst in all these tests. Additionally, the mean MMSE was 16.7 +/− 5.2 and was the lowest for AD compared to other patients. MMSE = 15 and below signifies the probability of total impairment, and there are such patients in this group. SVM and k-Nearest Neighbors (k-NN) algorithms were used to classify AD vs. control, bvFTD vs. control, and AD v. bvFTD, and they obtained a mean accuracy above 0.92. The svPPA group was too small for the ML procedures [81].

As we have described above (see Figure 3), there are many similarities in the properties of eye movements between AD and PD. However, in our projects, we have not only predicted the PD stage (UPDRS) from reflexive EM and neuropsychological test results, but also tried to predict the progression of the disease (longitudinal studies). We have also demonstrated that we can take a group of more advanced Parkinson’s disease patients as a ”Model”. This approach allows us to see PD progression as patients’ symptoms change relative to the model, which means that the multidimensional sets of their values (granules) become more similar to the set (granule) of the model. This approach allows us to compare different rates of various patients’ disease development and relate the effectiveness of different treatments (our different groups of PD: BMT, DBS, and POP).

Another critical issue is related to disease progression and related changes in different parts of the brain or symptoms related to the motor/cognitive and emotional systems [41]. We have estimated PD progression by all of our (general) attributes, by only motor-related attributes, and by only cognitive symptoms. The general and motor attributes predicted disease progression (UPDRS changes) well, but the correlation with cognitive attributes was much weaker [41]. The cognitive attributes do not change with disease progression in all PD patients, and this is the opposite of Alzheimer’s disease, which is mainly related to cognitive changes (MMSE changes).

We have demonstrated for PD patients that the depression score (measured by Beck’s depression inventory) is an essential attribute in the estimation of disease progression, and its value significantly increases the accuracy of UPDRS estimation [42]. In Alzheimer’s disease, depression is a significant factor, especially for older subjects with late onset of AD (LOAD) and for patients over 65 years of age [82]. Another essential function related to preclinical AD relates to motor symptoms, and they can even predict MCI [83]. They are related to muscle strength decrease and deficient grip strength [83]. Physical frailty, gait and balance problems, and loss of other motor functions can all precede cognitive impairment by several years. Even the trajectory of gait speed can precede MCI by 12 years [84,85]. Therefore, experiments in VR using familiar and novelty objects could also test the subject’s motor abilities and responses to different emotions.

There are two different ML classification approaches for predicting ND symptoms. The first one, more clinically oriented, is that based on neuropsychological and clinical tests, experienced neurologists decide each patient’s disease stage: (1) in Alzheimer’s disease—normal, MCI, dementia; (2) in Parkinson’s disease—normal, early-stage, medium, or advanced, or they can use the Hoehn and Yahr scale. In the next step, different parameters of EM, with the help of ML methods, attempt to classify patients following the doctors’ findings. In [80], the authors used five parameters related to novelty preference to differentiate normal and MCI subjects based on training normal subjects and AD patients. There were also five parameters related to RS, AS, and pursuit EM used to differentiate no disease from AD and AD from FTD (frontotemporal dementia) [81]. Our approach was different, as we took all eight neuropsychological and clinical parameters and four EM attributes together with ten to twelve parameters (in different studies) to classify four different ranges of PD stages. An even higher dimension of parameters was used in the BIBIOCARD study, where there were 181 condition parameters used to classify four stages of the disease—normal = 1, impaired and not MCI = 2, MCI = 3, dementia = 4—in a longitudinal study lasting over twenty-five years [86]. However, they have only used nine attributes, namely age, education, two cognitive testing results, two MRI scan results, two cerebrospinal fluid parameters, and APOE genotype, to predict the probability of progression from normal to MCI in the next 5 years. We have used their 11 cognitive test results and APOE genotype to determine, with the granular computing approach, that some of their normal patients might already have mild/very mild dementia or questionable impairment [87].

## 5. Conclusions

Alzheimer’s and Parkinson’s diseases are two of the most common neurodegenerative age-dependent diseases for which, despite many years of intensive research, we still do not have a cure. Both diseases have complex etiology and many years of hidden, non-reversible neurodegenerative processes (ND) with devastating effects on the brain. When the first symptoms appear, a large part of the brain has already disappeared. One possible solution is to find the neurodegenerative processes early enough to test possible methods to slow them down or even cure them.

We have described many papers showing that eye movement parameters change with disease progression and that they are also sensitive to the early stages of both diseases. However, the parameter space describing different measured attributes and methods has a very high dimension. To find significant subspace(s) with parameters sensitive to disease progression, we have proposed the use of different AI (machine learning) algorithms. These algorithms describe disease symptoms more precisely than the standard approach and can also predict disease development. We have given several examples of such AI (ML) methods and have demonstrated their effectiveness for the most common neurodegenerative diseases.

Therefore, the major differences between classical (statistical) and AI approaches are not only opportunities to reduce the dimension of the parameter space, but also to ask diverse questions about the nature of the diseases. For example, as mentioned above, many authors found pathological saccade parameters in AD and PD, but using the AI approach, we ask a different question: can EM parameters predict the AD/PD progression in different individual patients having different treatments, and in diverse disease stages? Responses to this question are related to the mechanisms of the disease.

There is a related difference between subject testing in real and virtual worlds. In the real environment, we can find many differences in behaviors such as EM, emotions, or peripheral body movements between ND and control subjects. We cannot control significant features of the real world and determine how changes in this environment may influence the subject. However, in virtual reality, we can ask the question: how do different features in the surroundings influence the individual subject? Again, finding isolated elements in the surroundings that, in a unique way, influence the behavior of the individual subjects gives insight into ND mechanisms. 

We think that the next research step should be into virtual reality. VR offers tremendous possibilities in the experimental environment. Researchers can simulate virtually every scenario in a strictly controlled environment where interactions can be controlled in real time. Furthermore, the patient is always safe because the experiment occurs in the laboratory or physician’s office. Development environments, such as Unity 3D, are offered free of charge and enable the design and development of VR applications. Therefore, virtual reality applications can become a valuable tool for dementia assessment if only more interdisciplinary teams, combining health professionals and computer scientists, would pave the way to new applications in this area. Much research has been done already, and more research is required on the scale of its usability for patients.

## Figures and Tables

**Figure 1 sensors-23-02145-f001:**
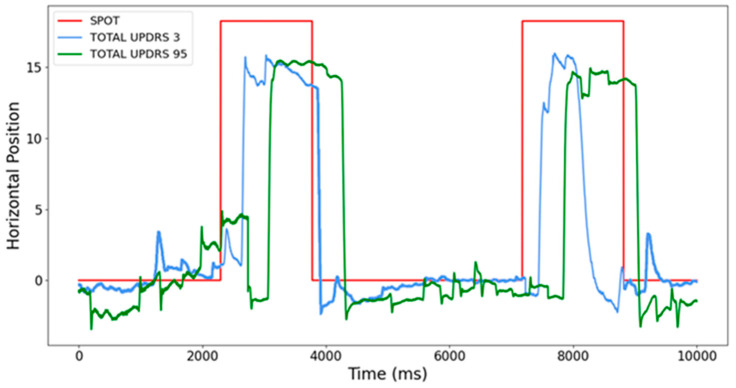
Saccades at various stages of PD. Recordings were performed with different UPDRS measured in clinical conditions under a doctor’s supervision. The red line is related to light spot movements, and the blue and green curves are related to reflexive saccades performed by two different PD patients in different stages of the disease: the blue curve denotes EM at the beginning of the disease, and the green curve at advanced PD. Notice the significant difference in response latencies [11].

**Figure 2 sensors-23-02145-f002:**
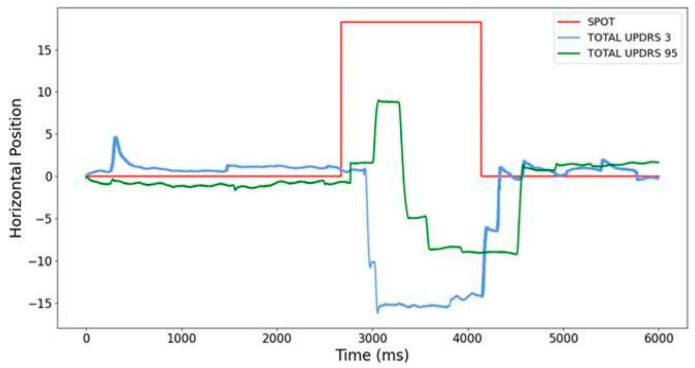
Antisaccades (AS) at various stages of PD. The red line is related to the light spot movements, and the blue and green curves are related to reflexive AS performed by two different PD patients in different stages of the disease: the blue curve denotes EM at the beginning of the disease, and the green curve in the advanced PD stage. Notice the significant difference not only in response latencies, but also in AS speed, and that a more advanced PD patient started with a saccade that changed to AS [21].

**Figure 3 sensors-23-02145-f003:**
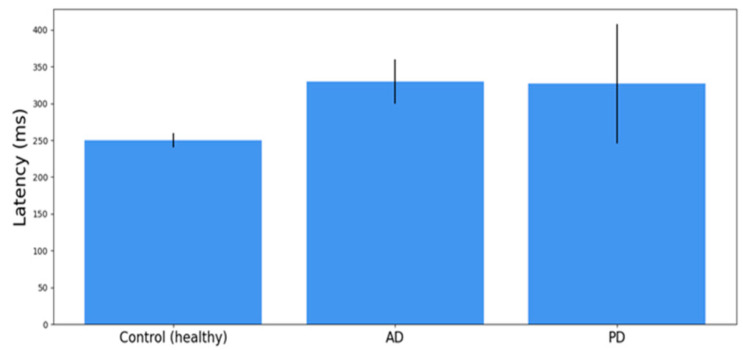
A comparison of the average results +/− SD of the AS latencies, obtained in a group of 27 age-matched normal controls (healthy people), 10 AD subjects, and 12 patients with PD. The averages of the control group and AD patients are derived from the research results obtained from [43] and the average of the PD patients from the results obtained from [11].

**Figure 4 sensors-23-02145-f004:**
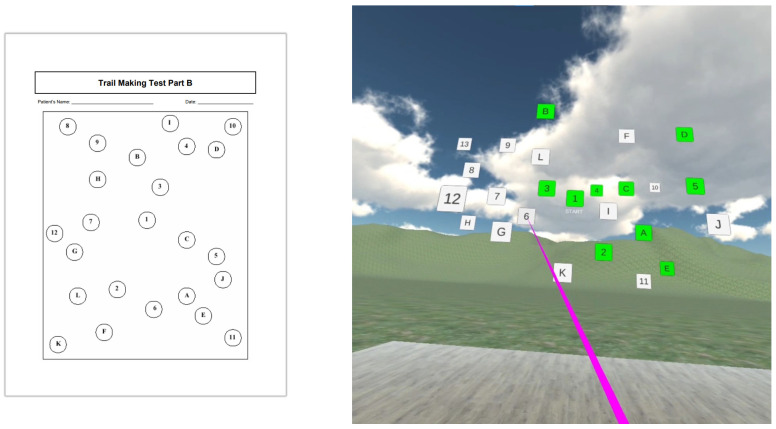
The prototype of an application testing the recognition of numbers and letters in the correct order implemented in a VR environment on an Oculus Go device. Trail B is generally thought to be sensitive to executive functioning since it requires a wide range of skills to complete. In 2D, it engages attention, memory, visual screening abilities, motor functioning, and cognitive processes, and it is likely to be even more difficult in 3D. Squares in green are already marked, and in white will be chosen in the proper order by the pointer in pink.

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
