# Peer review of "Machine Learning and Eye Movements Give Insights into Neurodegenerative Disease Mechanisms"

_sensors, 2023, doi:10.3390/s23042145_

Round 1

Reviewer 1 Report

The authors reviewed the results of psychological, neurological, and EM tests to predict progression of diseases such as AD and PD with Machine Learning methods. They concluded that EM parameters with clinical symptoms are powerful precision instruments and predictions of ND progression. The topic and conclusion are very interesting. This review would be needed minor additional explanations.

1.        The readers would not understand the machine learning methods, and how the methods were used and give the insights into ND mechanisms. Although the term “machine learning” was adopted in the title, the readers can read it only one in page 5. Please explain it in throughout the text.

2.        Insert a period: [5]. in P2.

3.        Insert a ward “a study”: a study [13]; [25]; [26].

Author Response

Dear Reviewer,

  1. Machine learning was added to abstract and in introduction
  2. corrected
  3. added

Reviewer 2 Report

The authors say that virtual reality applications can become a valuable tool for dementia assessment, but they don't talk about it neither in abstract nor in introduction section.

It is recommended to add the discussion of virtual reality in the abstract and in introduction.

It is recommended to discuss the combination of AI and Virtual reality and to analyze its use in the identification of Neurodegenerative Diseases.

Figure 4 is not readable.

Please mention all the figure in the text respectively.

Please extend the Conclusions part. 

Author Response

We have added a discussion about virtual reality in the abstract, introduction, and conclusions. We also have extended the conclusions part. 

All figures are now mentioned in the text.

We now have better described the Fig. 4, its right part is the copy from the computer screen, so it is difficult to change it.

Round 2

Reviewer 2 Report

The authors have edited the paper according the comments.